# Worker Size Diversity Has No Effect on Overwintering Success under Natural Conditions in the Ant *Temnothorax nylanderi*

**DOI:** 10.3390/insects12050379

**Published:** 2021-04-22

**Authors:** Romain Honorio, Claudie Doums, Mathieu Molet

**Affiliations:** 1Institute of Ecology and Environmental Sciences of Paris (IEES-Paris), Sorbonne Université, UPEC, CNRS, IRD, INRAE, 75005 Paris, France; mathieu.molet@sorbonne-universite.fr; 2Institut de Systématique, Evolution, Biodiversité (ISYEB), Muséum National d’Histoire Naturelle, CNRS, Sorbonne Université, EPHE-PSL, Université des Antilles, 75005 Paris, France; claudie.doums@ephe.psl.eu; 3Ecole Pratique des Hautes Etudes-Paris Sciences Lettre University, 75014 Paris, France

**Keywords:** social insects, mean body size, size distribution, survival, winter, colony size, worker demography

## Abstract

**Simple Summary:**

Winter is a harsh season for organisms living in temperate zones. Winter is often associated with starvation and cold temperatures, and these pressures can strongly affect organism survival. Living in groups can help these animals to cope with winter pressures. Social groups contain individuals which can vary in different ways: physiology, behavior, morphology, etc. In social insects such as ants, worker size leads to different responses to starvation and cold temperature in the laboratory. In this study, we investigated whether worker size affects colony and individual survival under natural conditions. We manipulated both worker size diversity and mean worker size within colonies of the ant *Temnothorax nylanderi*, reintroduced them in the field, and measured colony survival after overwintering. We found similar colony and individual (both adults and young) survival during winter between treatment colonies with reduced size diversity and/or manipulated mean worker size compared to control colonies with unmanipulated worker size. This result highlights that worker size diversity has no influence on colony performance in this species and more broadly questions the interest of worker size in social insect species with moderate worker size diversity. We discuss the potential sources of worker size diversity, including social context and selfish behavior.

**Abstract:**

Winter is a difficult period for animals that live in temperate zones. It can inflict high mortality or induce weight loss with potential consequences on performance during the growing season. Social groups include individuals of various ages and sizes. This diversity may improve the ability of groups to buffer winter disturbances such as starvation or cold temperature. Studies focusing on the buffering role of social traits such as mean size and diversity of group members under winter conditions are mainly performed in the laboratory and investigate the effect of starvation or cold separately. Here, we experimentally decreased worker size diversity and manipulated worker mean size within colonies in order to study the effect on overwintering survival in the ant *Temnothorax nylanderi*. Colonies were placed under natural conditions during winter. Colony survival was high during winter and similar in all treatments with no effect of worker size diversity and mean worker size. Higher brood survival was positively correlated with colony size (i.e., the number of workers). Our results show that the higher resistance of larger individuals against cold or starvation stresses observed in the laboratory does not directly translate into higher colony survival in the field. We discuss our results in the light of mechanisms that could explain the possible non-adaptive size diversity in social species.

## 1. Introduction

Winter is a season of resource scarcity, desiccation and cold temperatures that strongly affects the physiology of individuals. It causes drastic decreases in metabolic reserves [1,2], water content [3] and immune defenses [4,5]. To cope with winter, poikilotherm organisms such as insects have developed different strategies. They can either migrate to milder habitats, or produce specific winter-adapted individuals that differ from the summer individuals when the life cycle includes several generations, or overwinter [6]. Overwintering involves physiological modifications that lead to growth interruption or slowdown and to resistance to low temperatures, freezing, and other winter-associated constrains [6,7]. Overwintering also has indirect consequences on the future survival and reproductive success of individuals as it strongly affects the amount of energy available at the end of winter for the growing and reproductive seasons [1,8].

Social life could play a buffering role against harsh winter conditions. Animal societies can exhibit genetic, behavioral, physiological and morphological diversity among individuals of the same group and this is thought to broadly provide benefits to the group [9,10]. For instance, insect societies with more behaviorally and genetically diverse individuals have higher colony performance [11,12]. In social groups, the exchange of food (through trophallaxis [13,14,15,16,17]) or the role of brood as food reserve [14,18,19] can reduce the impact of starvation during hibernation. Several studies showed that individuals survive starvation better in groups [13,20,21]. For instance, workers in colonies with a mix of brood and adults survive starvation for a longer period than workers alone [18]. Workers within insect colonies can also show size diversity, which is often linked to a greater division of labor (e.g., [19,20,21]), but could also play a role against winter pressures. Larger individuals or workers with specific morphology can store food [22,23,24] and are known to survive longer under starvation in ants [18,25]. In addition to starvation, social life could also improve resistance to cold temperatures during overwintering. Elaborate nests (reviewed in [26]) and regulation of temperature with specific individual behaviors (reviewed in [27]) can buffer external temperature variation. For instance, individuals in colonies are able to form clusters to protect adults or offspring from cold temperatures [13,28]. The size of workers could also affect their resistance to temperature. In *Solenopsis invicta*, smaller individuals freeze at lower temperature [29] while in *Leptothorax acervorum*, larger workers survive longer under cold temperatures [25]. Even if size diversity of members in social groups can help these groups to cope with winter pressures, as mentioned above, hibernation can induce worker mortality [30,31,32] and even colony mortality in ants and honey bees [30,33,34,35].

To our knowledge, few studies have focused on the role of individual size and its diversity in the overwintering performance of social insects in complete and realistic environmental conditions. They are all based on founding queens of annual wasp and bee species, and they find that larger queens survive winter better, presumably because they have more metabolic reserves (wasps [36,37]—bees [38,39,40,41]). However, in perennial social insects such as ants, the whole colony hibernates together. This can increase survival by up to 190% compared to an isolated queen (in the laboratory [42]). Although previous studies showed that worker size diversity can improve resistance to starvation and cold temperatures as isolated stresses in laboratory experiments (e.g., [18,25,29,43,44,45,46,47,48,49,50,51]), it is still unclear whether it could provide colony-level benefits during realistic multifactorial overwintering conditions.

The ant *Temnothorax nylanderi* is an appropriate model to explore this topic because colonies nest above ground in hollow acorns and twigs and are thus exposed to the external weather conditions [52]. This species has moderate worker size diversity within colonies; that is, a lack of a soldier caste and a relatively low coefficient of variation (less than 0.06 [53,54]) compared to species with high worker size diversity (coefficient of variation higher than 0.3 or with discrete sub-castes such as soldier). In *T. nylanderi* and in a closely-related species (*Leptothorax acervorum*), larger workers survive longer when subject to starvation and cold [18,25]. In *Temnothorax* ants, overwintering in the field can lead to 50% queen mortality, and up to 70% worker mortality [30,32]—but see [55], where almost no mortality occurred. We manipulated both mean worker size and worker size diversity within colonies to disentangle the two effects. The study took place in a natural environment for several months to quantify the impact of worker size on colony survival (queen, worker and brood survival), and to test whether individual-level size-related resistances to cold and starvation observed in the laboratory transpose to colonies as a whole in the field. Recently, Honorio et al. (2020) manipulated worker size within colonies of this species in early spring. They found no effect on survival, growth and reproductive success in the field during the growing season. In contrast, during the harsh winter season, we hypothesized that decreasing worker mean size and diversity would decrease worker and colony survival. We also predicted that colonies with more workers or more brood would survive better because colony size buffers disturbances and larvae could be used as food resources.

## 2. Material and Methods

### 2.1. Study Model

One hundred and fifty colonies of the tiny acorn ant *Temnothorax nylanderi* were collected in October 2019 in the “Bois de Vincennes” forest (Paris, France, 48°50′22.14′′ N, 2°26′51.96′′ E). In the laboratory, each colony was transferred to an artificial nest consisting of two microscope slides separated by a 1-mm auto-adhesive plastic foam with three chambers. The nest was placed in a plastic box (11.5 cm × 11.5 cm × 5.5 cm) providing a foraging area. Colonies were kept for two weeks in a climatic chamber at 10–12 °C with a natural photoperiod. Water was provided ad libitum in a tube plugged with cotton. Workers and brood were counted.

### 2.2. Manipulation of Worker Size

Out of our 150 colonies, we excluded 31 queenless colonies, four polygynous colonies and seven colonies parasitized by a cestode [56]. We selected 80 colonies containing one queen and at least 70 workers (field colony size). Workers were counted under a binocular microscope. Our manipulation included four treatments that consisted of the removal of 50% of workers from colonies. Each treatment involved 20 colonies. In treatment 25S25L, we decreased worker diversity without changing mean worker size by removing the 25% smallest workers and the 25% largest workers. In treatments 50S and 50L, we decreased worker diversity but also respectively increased mean worker size by removing the 50% smallest workers or decreased mean worker size by removing the 50% largest workers. The last treatment was the control (treatment 50R) where we removed 50% of workers randomly. The worker removal protocols were similar to Colin et al. (2017) and Honorio et al. (2020) [53,54] and based on the apparent global body size: the sorting of large and small workers was done by eye under a stereoscopic microscope (Zeiss^®^, ×50 magnification) whereas the removal of random workers was done without a microscope to make sure that worker size could not be evaluated. This method was proven to significantly reduce the coefficient of variation of worker size by 30% in the treatment, while not altering it in the control [53]. We strictly followed this method that reliably decreases worker size diversity. We could not measure workers because such measurements must be performed on dead individuals, and our field setup did not allow us to recover corpses, in contrast with Colin et al. [53] whose setup took place in the laboratory. Colonies were assigned to the four treatments based on their number of workers in order to keep a similar distribution of colony sizes among the four treatments. Workers remaining in the colonies after manipulation and before the start of the experiment in the field constituted the initial colony size (69.1 workers ± 25.3 SD, min = 37, max = 141). Colonies initially contained 79.7 larvae ± 28.3 SD (min = 23, max = 170). By removing half of the colony workers to manipulate worker diversity without removing larvae, we initially decreased the worker to larva ratio (colony size/number of larvae) within colonies. The initial worker to larva ratio after manipulation and before the start of the experiment in the field was between 0.47 and 2.21 (mean = 0.90, median = 0.87, see Appendix A). Colonies were fed once with half a freshly killed mealworm (*Tenebrio molitor*) before reintroduction into the field.

### 2.3. Colony Rearing in the Field

We manufactured artificial nests to make them match natural nests as closely as possible, so that colonies could easily live in them after reintroduction in nature [54]. For that purpose, we used 2.5 cm × 2 cm (length × width) truncated cone corks. We drilled a 4, 5 or 6 mm-wide chamber from the larger side of the cone cork, and plugged this side with a glue gun to seal the gallery. Then, a 1 mm-wide entrance tunnel was pierced from the smaller side of the cone cork using a pointed plier in order to connect the chamber to the outside. In the laboratory, six corks (two of each size) were placed inside each plastic box, and we induced the emigration of colonies into the corks of their choice by removing the cover glass of the original nest. Then, the six corks from each plastic box (one containing the colony) were reintroduced in the “réserve ornithologique du Bois de Vincennes” in a semi-buried (10 cm deep) 40 cm × 35 cm (height × width) bucket with a pierced bottom (for water draining) and containing local soil. The bucket lid was cut off into a ring shape and the bottom side was covered with fluon^®^, a slippery coating, to prevent ant escape while retaining a wide entrance. The six corks were place randomly. This allowed colonies to relocate to the nest of their choice (size and location) inside the bucket whenever they wanted to. Indeed, *T. nylanderi* colonies readily switch nest depending on environmental conditions [57]. Because the 80 buckets had been in place for one and a half years, the soil and litter in the buckets was very similar to that of the surrounding forest; many arthropods and soil organisms could come in and out of the buckets. Colonies were left in the buckets from October 30th to March 16th, and subsequently collected and brought back to the laboratory. Corks were collected in the early morning, when workers were inactive because of cool temperatures, in order to make sure that complete colonies were collected. Cork nests were destroyed and colonies were forced to move to artificial microscope-slide nests. Workers and larvae were counted, and constituted respectively the final colony size and final larvae.

We considered that colonies had survived if they were recovered in March and if their queen was still alive. This loose definition of survival is an over-estimate relative to the proper life-history trait ‘colony survival’, as some colonies may actually have escaped from the bucket (entirely or only a fragment containing the queen, although this is highly unlikely because the bottom side of the lid was covered with a slippery coating—see Methods), and some colonies could have recovered from orphanage by later adopting a new fertilized queen [58]. Seven colonies lost their queen during overwintering (four 50S colonies, one 25L25S colony, one 50L colony and one 50R colony) and were excluded from the analyses of worker and larva survival. These colonies were mainly the ones that lost the most workers and larvae, suggesting imminent colony death. The rate of workers gain/loss was computed as: ((final colony size − initial colony size)/initial colony size) × 100. The rate of larvae gain/loss was computed as: ((final larvae − initial larvae)/initial larvae) × 100. In case of workers/larvae gain, these rates are positive; in case of workers/larvae loss, these rates are negative.

### 2.4. Statistical Analysis

We investigated the effect of our manipulation (predictor variable, four levels: 25S25L, 50S, 50L, 50R) on three dependent variables: colony survival, rate of workers gain/loss and rate of larvae gain/loss. In addition to treatment, we considered two other predictor variables: the initial colony size and the initial worker to larva ratio. Because of strong collinearity (Variance inflation factor > 5 [59]), we did not add the predictor variable ‘initial larvae’ which was highly correlated with ‘initial colony size’ (Spearman correlation, *r_s_* = 0.76, *p* < 0.001). We checked that our results were not affected by the choice of the retained variable by performing all analyses with the variables ‘initial larvae’ or ‘sum of initial larvae and initial colony size instead of ‘initial colony size’. The interaction between treatment and initial colony size was also included, to investigate potentially distinct responses associated with colony size. Large colonies could be more affected by the loss of worker size diversity than small colonies. Regarding rate of larvae gain/loss, we included the rate of workers gain/loss as a predictor variable because it reflected the loss of workforce available for larval rearing at the end of the experiment. The two models are summarized in Table 1.

All statistical analyses were carried out with R v3.6.1 (www.r-project.org, 30 January 2021). We performed a Fisher’s exact test (for small sample size) to compare colony survival among treatments. For other traits (rate of workers gain/loss and rate of larvae gain/loss), we used generalized linear models (GLMs), with Gaussian distribution (by visually checking data distribution). Normality of the residuals and homogeneity of variances were checked visually following [60]; no transformation of the data was necessary. For each analysis, the minimum adequate model was selected using a backward stepwise approach where predictor variables were removed one by one from a full model based on a log likelihood ratio test. We used log likelihood ratio tests to obtain the *p-*values for each predictor variable by comparing the minimum adequate model with a model excluding or including the variable of interest (depending on whether the variable was present in or absent from the minimal adequate model respectively). All plots were generated using ggplot2 [61]. Statistical power analyses are presented in the Appendix A based on the effect size observed in the results. Basically, we had the largest effect size, we had a high power to reject the null hypothesis, so that we can confidently conclude that here was no significant effect of our treatment. When the power was low, it was associated with a low effect size that would not be biologically relevant.

## 3. Results

We recovered 78 colonies out of 80 at the end of the experiment; the corks of the two missing colonies (50L) were found out of the bucket (probably projected outside by an animal). Seven colonies lost their queen during overwintering (see Material and Methods for details). Colony survival was therefore very high (71 colonies) and did not differ among treatments (Fisher’s exact test, *p* = 0.45). The seven queenless colonies were excluded from the statistical analyses (analyzing data with queenless colonies did not change the qualitative results presented in the manuscript). Our dataset for subsequent analyses thus consisted of 71 queenright colonies. Colonies lost on average 4.4% (±20.2% sd) of workers. Quite surprisingly, 39% of queenright colonies gained workers (with five colonies gaining more than 20% workers). The rate of workers gain/loss did not differ among treatments (*F*_70;67_ = 0.87, *p* = 0.46; Figure 1a) and was not explained by any predictor variable (Table 1). Colonies gained on average 0.53% (±32.0%) of larvae. The rate of larvae gain/loss increased with the initial worker to larva ratio, meaning that colonies with a higher number of workers relative to larvae managed to increase larva survival (*F*_68;69_ = 9.64, *p* = 0.003; Figure 2a). The rate of larvae gain/loss also increased with the rate of workers gain/loss, meaning that colonies with higher worker survival also had higher larva survival (*F*_68;69_ = 70.95, *p* < 0.001; Figure 2b). The rate of larvae gain/loss did not differ among treatments (*F*_68;65_ = 0.48, *p* = 0.69; Figure 1b). Statistics are detailed in Table 1.

## 4. Discussion

In this study, we experimentally manipulated mean worker size and worker size diversity within colonies of *Temnothorax nylanderi* to quantify the consequences on colony and worker survival during winter in the field. After overwintering, we found a high colony survival rate (91% were recovered and still queenright after overwintering) and a high worker survival (96% on average) in all treatments. Subsequently, we found no effect of the manipulation on colony survival and on the rate of workers gain/loss and larvae, highlighting that worker size (mean and diversity) is not a key component to get through winter. However, we found a positive relationship between the rate of larvae gain/loss with the initial worker to larva ratio and with the rate of workers gain/loss. This suggests that brood survival is mostly determined by the workforce available rather than worker size and diversity. In contrast, worker survival was not correlated with initial colony size during winter.

The lack of an effect of worker size (mean size and diversity) on overwintering performance is consistent with two other studies performed in the same species. The first used the same protocol but was performed during the growing season and found no impact on colony survival, growth or reproductive success during the growing season [54]. The second tested colony performance for various traits in the laboratory and also found no difference for all traits measured between colonies with unmanipulated or reduced worker size diversity [53]. Together, our findings and previous studies suggest that worker size (diversity and mean) had no effect on colony survival.

A positive effect of colony size on colony performance is often observed in social insect species [57,58,62,63]. In our study, larger colony size was beneficial for larval survival but not for worker survival or colony survival. The absence of a positive effect of colony size on worker survival is somewhat surprising given that it was linked to colony growth and reproductive success in this species [54]. The lack of a relationship between the colony size and colony survival during winter is also found in other studies on *Temnothorax* [14,30,64]. The high survival rate of both queens and workers in our study during winter is unexpected. Two previous studies in the field on *Temnothorax* species reported relatively high queen and worker mortality [30,32]. However, Mitrus [55] found a very low variation rate of workers during winter with almost no queen mortality. Overwintering mortality could vary greatly over the years depending on how harsh winter is, because *T. nylanderi* colonies nest above ground and directly face external temperatures [30]. During our experiment, winter was particularly mild (3 °C above average, although temperatures still dropped between 0 °C and −2 °C on eight nights; source: Météo France). This could explain the low mortality. However, high winter temperatures are not necessarily beneficial for overwintering colonies. Indeed, they could cause an increase in the consumption of metabolic reserves, leading to their early depletion and subsequent mortality in ants [31,65], more generally in insects [66] and terrestrial organisms [8]. Nevertheless, overwintering above ground enables *Temnothorax* colonies to become active earlier and to be better prepared for spring activity [67].

Quite surprisingly, we found that some colonies grew over winter. Our field devices prevent the escape of ants from the experimental colonies within the buckets but they remain accessible to ants coming from the outside. A few workers were observed on the bucket lids during the experiment. Ants from foreign colonies could thus enter our experimental colonies. Worker drifting and colony fusion are very common processes in *T. nylanderi*, especially in winter, because of low nest availability [58,68,69]; this is also observed in various social insects (wasps [70]—bumble bees [71]—honey bees [72]). Although increases in colony size after overwintering could result from the production of new workers in *T. Crassispinus* as Mitrus (2015) [55] suggested, we discard this hypothesis because (1) brood emergence into adults is a highly synchronized summer process in *T. nylanderi* [54,73,74] (2) our colonies contained no nymphs neither before nor after overwintering; and (3) Penick et al. [75] showed that higher temperatures are required for brood development in *Temnothorax*.

Overall, the higher resistance of larger individuals against starvation or cold temperature observed in the laboratory (respectively in *T. nylanderi* and *L. acervorum* [18,25]) did not transpose to the colony level in the field, highlighting the need to directly evaluate the adaptive value on colony-level traits. Our winter study, coupled with Honorio et al.’s growing season study [54], further questions the role of worker size (mean and diversity) in *T. nylanderi*, and more generally in ant species with moderate worker size diversity. Surprisingly, neither mean worker size nor worker size diversity improve division of labor during the growing season or confer higher resistance to harsher conditions in winter. In some social insect species, like bumble bees, worker size diversity does not improve colony performance [76,77] whereas mean worker size does [78]. More studies disentangling the effects of mean size and size diversity are needed in various social species, to investigate the role of each component (mean and diversity) in colony performance. Worker size within colonies can result from different factors, either intrinsic (genetic or developmental) or external (food, temperature, ecological constraints) (reviewed in [79]). More broadly, the social environment can modulate the produced phenotypes by controlling the influence of each external factors (e.g., [80]). As discussed by Honorio et al. [54], in the absence of clear adaptive value as in *T. nylanderi*, size diversity could result either from selfish larval attempts at developing into larger individuals with developed ovaries [81] to get higher inclusive fitness or from low developmental canalization in a social context [82,83]. Unlike solitary life, social life could buffer external disturbances (e.g., [84,85,86]) and thus reduce the selective pressures on individuals within the society, leading to size variation within colony without any group-level benefits.

## Figures and Tables

**Figure 1 insects-12-00379-f001:**
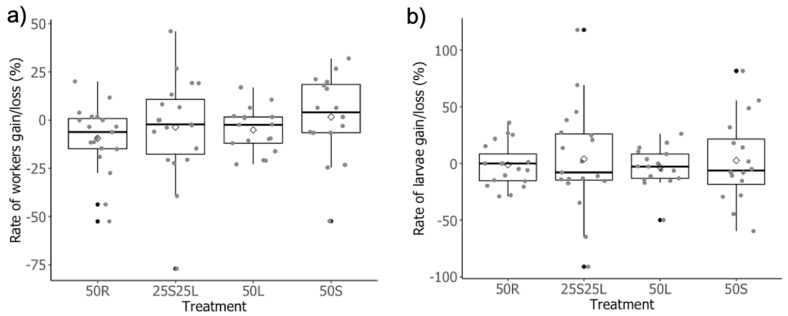
Boxplots comparing the effects of manipulation (four levels) on (**a**) the rate of workers gain/loss and (**b**) the rate of larvae gain/loss. 50R: random removal of 50% of workers; 25S25L: removal of the 25% smallest and the 25% largest workers; 50L: removal of the 50% largest workers; 50S: removal of the 50% smallest workers. Twenty colonies were assigned to each treatment at the beginning of the experiment. Boxes show median, quartiles and extremes (black circles). Mean is represented by empty diamond. Raw data are shown by grey circles. Treatment had no effect. Statistics are presented in Table 1.

**Figure 2 insects-12-00379-f002:**
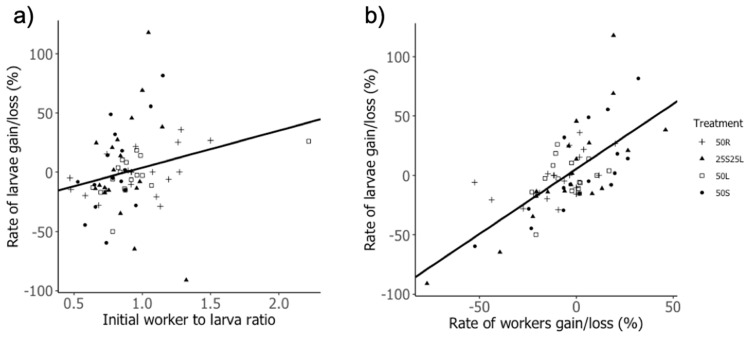
Relationship between the rate of larvae gain/loss and two social traits. The rate of larvae gain/loss increased with (**a**) the initial worker to larva ratio (intercept = −27.59, slope = 31.35) as well as (**b**) the rate of workers gain/loss (intercept = 5.35, slope = 1.09). The regression lines are drawn from the coefficients of the model. Statistics are presented in Table 1.

**Table 1 insects-12-00379-t001:** Models and statistics related to the rate of workers gain/loss and the rate of larvae gain/loss. ‘Minimum model’ means that the predictor was retained in the selected minimum model. The minimum model for the dependent variable ‘Rate of workers gain/loss’ was the null model. Significant effects are shown in bold.

Predictor Variables	Dependent Variables	
	Rate of Workers Gain/Loss	Rate of Larvae Gain/Loss
Treatment	*F*_70;67_ = 0.87, *p* = 0.46	*F*_68;65_ = 0.48, *p* = 0.69
Initial colony size	*F*_70;69_ = 0.10, *p* = 0.75	*F*_68;67_ = 0.06, *p* = 0.80
Treatment—initial colony size interaction	*F*_70;66_ = 0.92, *p* = 0.45	*F*_68;64_ = 0.35, *p* = 0.84
Initial worker/larva ratio	*F*_70;69_ = 0.002, *p* = 0.96	**Minimum model** ***F*_68;69_ = 9.64,** ***p* = 0.003**
Rate of workers gain/loss		**Minimum model** ***F*_68;69_ = 70.95,** ***p* < 0.001**

## Data Availability

The dataset analyzed during the current study and R scripts are available on zenodo (10.5281/zenodo.4686757).

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
