# Peer review of "Worker Size Diversity Has No Effect on Overwintering Success under Natural Conditions in the Ant Temnothorax nylanderi"

_insects, 2021, doi:10.3390/insects12050379_

Round 1

Reviewer 1 Report

In this study, the authors study the effect of worker size on colony survival in the field in the ant Temnothorax nylanderi. To do so, they experimentally manipulate worker size diversity, as well as the average worker size, of colonies that they reintroduced in the field for the winter season. After winter, they bring the colonies back to the lab and measure the effect of their manipulations on queen, worker and larva survival.

I think this is a very nice study. The main strength in my opinion is the reintroduction of the colonies in the field to test the effect of overwintering, as laboratory studies cannot mimic all the factors experienced by field colonies (fluctuations in temperature and humidity, light/dark rhythm, etc).

The fact that the authors did not detect an effect of overwintering (in other words, that they report “negative results”) should not be an obstacle for publication, but it does require some additions to the manuscript in my opinion.

First, the authors should acknowledge some drawbacks of their experimental manipulations. One concern I have is that they use very specific numbers to describe their manipulations (e.g., top and bottom 25%) despite none of the workers having been measured (if I understand correctly). The authors simply gauged by eyes which workers were larger, and which were smaller (which may be difficult as, according to them, the variation is rather minimal in this species). Using specific proportions in the description of the methods is misleading, and unlikely to be true. I think the authors should phrase their manipulations differently and more accurately (e.g., removal of large and small individuals), to better represent what they actually did. As it is, the descriptions suggest a level of precision that was not achieved in this study.

Another, related concern, is that the authors did not measure the effect of their manipulations (I now realize that I don’t think they measured a single worker in a study about worker size…). They merely rely on published work to support their claim that worker size was indeed different after manipulations. This issue, together with the absence of measurements to choose which workers to keep and which to remove, makes me doubt of the efficiency of their manipulations. This is important because inefficient manipulations are a valid alternative explanation for not finding an effect on colony survival. If I am not mistaken and I have understood the methods correctly, these drawbacks should be clearly acknowledged in the discussion of the paper, and proposed as possible explanations of their results.

The study has a relatively high number of biological replicates, but I think a power analysis to estimate the likelihood of detecting an effect if there was one would strengthen the conclusion.

I am surprised that the authors do not plan to make the data accessible with the manuscript upon publication (“available from the corresponding author on reasonable request”). In my opinion, the data should be uploaded to a data repository or provided as a supplement to be freely accessible in the future. As papers become older, it is more and more difficult to contact corresponding authors, and this should not go in the way of reproducible science. I would also recommend that the authors upload their annotated R scripts to allow their peers to reproduce their data analyses.

In the introduction, the authors could provide more background on studies of the link between worker diversity (not only in term of size, but also genetic background, age, etc), division of labor and colony performance.

L21: should be “compared to”.

L38: “non-adaptive” is a bit strong, it could be adaptive in other contexts.

L57-58: “survive starvation for a longer period”, compared to what?

L86: should be “a soldier caste”.

L757-76: should be “larva number”.

L196-197: I think a GLM with Gaussian distribution is merely a LM, isn’t it?

L422-423: Please check the format of references.

Reviewer 2 Report

I found your paper to be well executed - generally well-written, a good experimental design, and thorough analysis.  The basic premise was a good one (that worker size diversity might influence overwintering success) that unfortunately led to mostly negative results.  We really need more papers like this to be published - where the basic scientific methodology is sound - even if the end result is not what the authors might have hoped for.  Your paper had a number of strengths worth celebrating: a good design and an experiment that took place in a natural context (unlike so many similar experiments that take place in a lab).  Part of learning what the true benefits of worker polymorphism are for social insect colonies means finding cases where such diversity doesn't play a role.  Even though your basic hypothesis was disproven the end results are informative.

Author Response

Reviewer 2 did not suggest any corrections.

Reviewer 3 Report

In this manuscript, the authors test whether the diversity of worker sizes produced within a social insect colony affects overwintering success. Social insects display a large range of body sizes, even within the nest. While some species exhibit discrete morphological castes, most display a continuous size variation among the workers within a colony. This continuous size distribution is often assumed to be adaptive, though this has been the focus of many studies. However, most of these studies occur in the lab, and there is a need for more field-based studies, which is an important contribution of this paper. Overall, I found the study interesting, especially since there was no relationship between the average size or distribution of worker sizes and colony overwintering success. Additionally, the paper is well written, and the data is presented nicely.

Comments:

Line 123: How was apparent global body size determine. Where all workers measured first then from these data small and large workers determined, or was the previous literature used, or was it more haphazard?

Line 136: I think it would be helpful to see the initial colony size and worker to larva ratio by treatment in the supplemental materials.

Line 170: Colony mortality may be skewed to colonies with smaller average size? Since overwinter survival is the paper's main focus, have you thought of including the excluded queenless colonies' data into your analyses? Would this change the results?

Figure 1: It appears as if colonies with a more than normal number of small workers, 25S25L and 50S, have more variation in their overwinter survival. I don’t know if this is statistically significant, but if it is, it would be an interesting result.

Line 270 – 278: Do you think the buckets used to house the ants increased overwintering success by buffering colonies from external environmental conditions or preventing nest destruction from predators or accidents?

Line 279 – 291: You hypothesize that the workers grew over winter due to ants from foreign colonies entering your experimental colonies. I appreciate your honesty that these events could have occurred. Do you think this influenced your results though? One reason you do not see any difference in the rate of workers gain/lost based on colony size, as mentioned in line 260 to be surprised by, may be due to these additional workers adding noise into the data. Also, do you have any idea the size of these workers?
